# COMPLEX LOGICAL REASONING OVER KNOWLEDGE GRAPHS USING LARGE LANGUAGE MODELS

## ABSTRACT

Reasoning over knowledge graphs (KGs) is a challenging task that requires a deep understanding of the complex relationships between entities and the underlying logic of their relations. Current approaches rely on learning geometries to embed entities in vector space for logical query operations, but they suffer from subpar performance on complex queries and dataset-specific representations. In this paper, we propose a novel decoupled approach, Language-guided Abstract Reasoning over Knowledge graphs (LARK), that formulates complex KG reasoning as a combination of contextual KG search and logical query reasoning, to leverage the strengths of graph extraction algorithms and large language models (LLM), respectively. Our experiments demonstrate that the proposed approach outperforms state-of-the-art KG reasoning methods on standard benchmark datasets across several logical query constructs, with significant performance gain for queries of higher complexity. Furthermore, we show that the performance of our approach improves proportionally to the increase in size of the underlying LLM, enabling the integration of the latest advancements in LLMs for logical reasoning over KGs. Our work presents a new direction for addressing the challenges of complex KG reasoning and paves the way for future research in this area.

## 1 INTRODUCTION

Knowledge graphs (KGs) encode knowledge in a flexible triplet schema where two entity nodes are connected by relational edges. However, several real-world KGs, such as Freebase (Bollacker et al., 2008), Yago (Suchanek et al., 2007), and NELL (Carlson et al., 2010), are often large-scale, noisy, and incomplete. Thus, reasoning over such KGs is a fundamental and challenging problem in AI research. The over-arching goal of logical reasoning is to develop answering mechanisms for first-order logic (FOL) queries over KGs using the operators of existential quantification ($\exists$), conjunction ($\wedge$), disjunction ($\vee$), and negation ($\neg$). Current research on this topic primarily focuses on the creation of diverse latent space geometries, such as vectors (Hamilton et al., 2018), boxes (Ren et al., 2020), hyperboloids (Choudhary et al., 2021b), and probabilistic distributions (Ren and Leskovec, 2020), in order to effectively capture the semantic position and logical coverage of knowledge graph entities. Despite their success, these approaches are limited in their performance due to the following. (i) *Complex queries*: They rely on constrained formulations of FOL queries that lose information on complex queries that require chain reasoning (Choudhary et al., 2021a) and involve multiple relationships between entities in the KG, (ii) *Generalizability*: optimization for a particular KG may not generalize to other KGs which limits the applicability of these approaches in real-world scenarios where KGs can vary widely in terms of their structure and content, and (iii) *Scalability*: intensive training times that limit the scalability of these approaches to larger KGs and incorporation of new data into existing KGs. To address these limitations, we aim to leverage the reasoning abilities of large language models (LLMs) in a novel framework, shown in Figure 1, called Language-guided Abstract Reasoning over Knowledge graphs (LARK).

In LARK, we utilize the logical queries to search for relevant subgraph contexts over knowledge graphs and perform chain reasoning over these contexts using logically-decomposed LLM prompts. To achieve this, we first abstract out the logical information from both the input query and the KG. Given the invariant nature of logic[1], this enables our method to focus on the logical formulation,

---

[1]logical queries follow the same set of rules and procedures irrespective of the KG context.

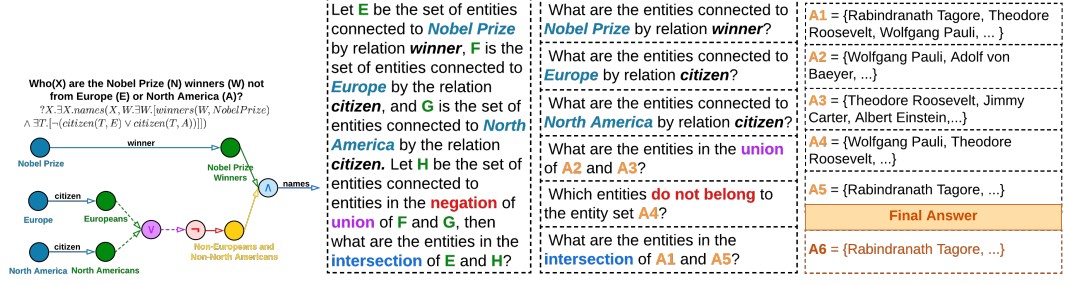

Figure 1: Example of LARK's query chain decomposition and logically-ordered LLM answering for effective performance. LLMs are more adept at answering simple queries, and hence, we decompose the multi-operation complex logical query (a,b) into elementary queries with single operation (c) and then use a sequential LLM-based answering method to output the final answer (d).

avoid model hallucination[2], and generalize over different knowledge graphs. From this abstract KG, we extract relevant subgraphs using the entities and relations present in the logical query. These subgraphs serve as context prompts for input to LLMs. In the next phase, we need to effectively handle complex reasoning queries. From previous works (Zhou et al., 2023; Khot et al., 2023), we realize that LLMs are significantly less effective on complex prompts, when compared to a sequence of simpler prompts. Thus to simplify the query, we exploit their logical nature and deterministically decompose the multi-operation query into logically-ordered elementary queries, each containing a single operation (depicted in the transition from Figure 1b to 1c). Each of these decomposed logical queries is then converted to a prompt and processed through the LLM to generate the final set of answers (shown in Figure 1d). The logical queries are handled sequentially, and if query $y$ depends on query $x$, then $x$ is scheduled before $y$. Operations are scheduled in a logically-ordered manner to enable batching different logical queries together, and answers are stored in caches for easy access.

The proposed approach effectively integrates logical reasoning over knowledge graphs with the capabilities of LLMs, and to the best of our knowledge, is the first of its kind. Unlike previous approaches that rely on constrained formulations of first-order logic (FOL) queries, our approach utilizes logically-decomposed LLM prompts to enable chain reasoning over subgraphs retrieved from knowledge graphs, allowing us to efficiently leverage the reasoning ability of LLMs. Our KG search model is inspired by retrieval-augmented techniques (Chen et al., 2022) but realizes the deterministic nature of knowledge graphs to simplify the retrieval of relevant subgraphs. Moreover, compared to other prompting methods (Wei et al., 2022; Zhou et al., 2023; Khot et al., 2023), our chain decomposition technique enhances the reasoning capabilities in knowledge graphs by leveraging the underlying chain of logical operations in complex queries, and by utilizing preceding answers amidst successive queries in a logically-ordered manner. To summarize, the primary contributions of this paper are as follows:

1. We propose, Language-guided Abstract Reasoning over Knowledge graphs (LARK), a novel model that utilizes the reasoning abilities of large language models to efficiently answer FOL queries over knowledge graphs.
2. Our model uses entities and relations in queries to find pertinent subgraph contexts within abstract knowledge graphs, and then, performs chain reasoning over these contexts using LLM prompts of decomposed logical queries.
3. Our experiments on logical reasoning across standard KG datasets demonstrate that LARK outperforms the previous state-of-the-art approaches by $35\% - 84\%$ MRR on 14 FOL query types based on the operations of projection (p), intersection ($\wedge$), union ($\vee$), and negation ($\neg$).
4. We establish the advantages of chain decomposition by showing that LARK performs $20\% - 33\%$ better on decomposed logical queries when compared to complex queries on the task of logical reasoning. Additionally, our analysis of LLMs shows the significant contribution of increasing scale and better design of underlying LLMs to the performance of LARK.

---

[2]the model ignores semantic common-sense knowledge and infers only from the KG entities for answers.

## 2 RELATED WORK

Our work is at the intersection of two topics, namely, logical reasoning over knowledge graphs and reasoning prompt techniques in LLMs.

**Logical Reasoning over KGs:** Initial approaches in this area (Bordes et al., 2013; Nickel et al., 2011; Das et al., 2017; Hamilton et al., 2018) focused on capturing the semantic information of entities and the relational operations involved in the projection between them. However, further research in the area revealed a need for new geometries to encode the spatial and hierarchical information present in the knowledge graphs. To tackle this issue, models such as Query2Box (Ren et al., 2020), HypE (Choudhary et al., 2021b), PERM (Choudhary et al., 2021a), and BetaE (Ren and Leskovec, 2020) encoded the entities and relations as boxes, hyperboloids, Gaussian distributions, and beta distributions, respectively. Additionally, approaches such as CQD (Arakelyan et al., 2021) have focused on improving the performance of complex reasoning tasks through the answer composition of simple intermediate queries. In another line of research, HamQA (Dong et al., 2023) and QA-GNN (Yasunaga et al., 2021) have developed question-answering techniques that use knowledge graph neighborhoods to enhance the overall performance. We notice that previous approaches in this area have focused on enhancing KG representations for logical reasoning. Contrary to these existing methods, our work provides a systematic framework that leverages the reasoning ability of LLMs and tailors them toward the problem of logical reasoning over knowledge graphs.

**Reasoning prompts in LLMs:** Recent studies have shown that LLMs can learn various NLP tasks with just context prompts (Brown et al., 2020). Furthermore, LLMs have been successfully applied to multi-step reasoning tasks by providing intermediate reasoning steps, also known as Chain-of-Thought (Wei et al., 2022; Chowdhery et al., 2022), needed to arrive at an answer. Alternatively, certain studies have composed multiple LLMs or LLMs with symbolic functions to perform multi-step reasoning (Jung et al., 2022; Creswell et al., 2023), with a pre-defined decomposition structure. More recent studies such as least-to-most (Zhou et al., 2023), successive (Dua et al., 2022) and decomposed (Khot et al., 2023) prompting strategies divide a complex prompt into sub-prompts and answer them sequentially for effective performance. While this line of work is close to our approach, they do not utilize previous answers to inform successive queries. LARK is unique due to its ability to utilize logical structure in the chain decomposition mechanism, augmentation of retrieved knowledge graph neighborhood, and multi-phase answering structure that incorporates preceding LLM answers amidst successive queries.

## 3 METHODOLOGY

In this section, we will describe the problem setup of logical reasoning over knowledge graphs, and describe the various components of our model.

### 3.1 PROBLEM FORMULATION

In this work, we tackle the problem of logical reasoning over knowledge graphs (KGs) $\mathcal{G} : E \times R$ that store entities ($E$) and relations ($R$). Without loss of generality, KGs can also be organized as a set of triplets $\langle e_1, r, e_2 \rangle \subseteq \mathcal{G}$, where each relation $r \in R$ is a Boolean function $r : E \times E \rightarrow \{True, False\}$ that indicates whether the relation $r$ exists between the pair of entities $(e_1, e_2) \in E$. We consider four fundamental first-order logical (FOL) operations: projection (p), intersection ($\wedge$), union ($\vee$), and negation ($\neg$) to query the KG. These operations are defined as follows:

$$q_p[Q_p] \triangleq ?V_p : \{v_1, v_2, ..., v_k\} \subseteq E \ \exists \ a_1 \tag{1}$$

$$q_\wedge[Q_\wedge] \triangleq ?V_\wedge : \{v_1, v_2, ..., v_k\} \subseteq E \ \exists \ a_1 \wedge a_2 \wedge ... \wedge a_i \tag{2}$$

$$q_\vee[Q_\vee] \triangleq ?V_\vee : \{v_1, v_2, ..., v_k\} \subseteq E \ \exists \ a_1 \vee a_2 \vee ... \vee a_i \tag{3}$$

$$q_\neg[Q_\neg] \triangleq ?V_\neg : \{v_1, v_2, ..., v_k\} \subseteq E \ \exists \ \neg a_1 \tag{4}$$

where $Q_p, Q_\neg = (e_1, r_1); \ Q_\wedge, Q_\vee = \{(e_1, r_1), (e_2, r_2), ..., (e_i, r_i)\}; \text{ and } a_i = r_i(e_i, v_i)$

where $q_p, q_\wedge, q_\vee$, and $q_\neg$ are projection, intersection, union, and negation queries, respectively; and $V_p, V_\wedge, V_\vee$ and $V_\neg$ are the corresponding results of those queries (Arakelyan et al., 2021; Choudhary et al., 2021a). $a_i$ is a Boolean indicator which will be 1 if $e_i$ is connected to $v_i$ by relation $r_i$, 0

otherwise. The goal of logical reasoning is to formulate the operations such that for a given query $q_\tau$ of query type $\tau$ with inputs $Q_\tau$, we are able to efficiently retrieve $V_\tau$ from entity set $E$, e.g., for a projection query $q_p[(\text{Nobel Prize, winners})]$, we want to retrieve $V_p = \{\text{Nobel Prize winners}\} \subseteq E$.

In conventional methods for logical reasoning, the query operations were typically expressed through a geometric function. For example, the intersection of queries was represented as an intersection of box representations in Query2Box (Ren et al., 2020). However, in our proposed approach, LARK, we leverage the advanced reasoning capabilities of Language Models (LLMs) and prioritize efficient decomposition of logical chains within the query to enhance performance. This novel strategy seeks to overcome the limitations of traditional methods by harnessing the power of LLMs in reasoning over KGs.

## 3.2 NEIGHBORHOOD RETRIEVAL AND LOGICAL CHAIN DECOMPOSITION

The foundation of LARK's reasoning capability is built on large language models. Nevertheless, the limited input length of LLMs restricts their ability to process the entirety of a knowledge graph. Furthermore, while the set of entities and relations within a knowledge graph is unique, the reasoning behind logical operations remains universal. Therefore, we specifically tailor the LLM prompts to account for the above distinctive characteristics of logical reasoning over knowledge graphs. To address this need, we adopt a two-step process:

1. **Query Abstraction:** In order to make the process of logical reasoning over knowledge graphs more generalizable to different datasets, we propose to replace all the entities and relations in the knowledge graph and queries with a unique ID. This approach offers three significant advantages. Firstly, it reduces the number of tokens in the query, leading to improved LLM efficiency. Secondly, it allows us to solely utilize the reasoning ability of the language model, without relying on any external common sense knowledge of the underlying LLM. By avoiding the use of common sense knowledge, our approach mitigates the potential for model hallucination (which may lead to the generation of answers that are not supported by the KG). Finally, it removes any KG-specific information, thereby ensuring that the process remains generalizable to different datasets. While this may intuitively seem to result in a loss of information, our empirical findings, presented in Section 4.4, indicate that the impact on the overall performance is negligible.

2. **Neighborhood Retrieval:** In order to effectively answer logical queries, it is not necessary for the LLM to have access to the entire knowledge graph. Instead, the relevant neighborhoods containing the answers can be identified. Previous approaches (Guu et al., 2020; Chen et al., 2022) have focused on semantic retrieval for web documents. However, we note that logical queries are deterministic in nature, and thus we perform a $k$-level depth-first traversal[3] over the entities and relations present in the query. Let $E_\tau^1$ and $R_\tau^1$ denote the set of entities and relations in query $Q_\tau$ for a query type $\tau$, respectively. Then, the $k$-level neighborhood of query $q_\tau$ is defined by $\mathcal{N}_k(q_\tau[Q_\tau])$ as:

$$\mathcal{N}_1(q_\tau[Q_\tau]) = \left\{(h, r, t) : \left(h \in E_\tau^1\right), \left(r \in R_\tau^1\right), \left(t \in E_\tau^1\right)\right\} \tag{5}$$

$$E_\tau^k = \{h, t : (h, r, t) \in \mathcal{N}_{k-1}(q_\tau[Q_\tau]\}, \quad R_\tau^k = \{r : (h, r, t) \in \mathcal{N}_{k-1}(q_\tau[Q_\tau]\} \tag{6}$$

$$\mathcal{N}_k(q_\tau[Q_\tau]) = \left\{(h, r, t) : \left(h \in E_\tau^k\right), \left(r \in R_\tau^k\right), \left(t \in E_\tau^k\right)\right\} \tag{7}$$

We have taken steps to make our approach more generalizable and efficient by abstracting the query and limiting input context for LLMs. However, the complexity of a query still remains a concern. The complexity of a query type $\tau$, denoted by $\mathcal{O}(q_\tau)$, is determined by the number of entities and relations it involves, i.e., $\mathcal{O}(q_\tau) \propto |E_\tau| + |R_\tau|$. In other words, the size of the query in terms of its constituent elements is a key factor in determining its computational complexity. This observation is particularly relevant in the context of LLMs, as previous studies have shown that their performance tends to decrease as the complexity of the queries they handle increases (Khot et al., 2023). To address this, we propose a **logical query chain decomposition mechanism** in LARK which *reduces a complex multi-operation query to multiple single-operation queries*. Due to the exhaustive set of operations, we apply the following strategy for decomposing the various query types:

- Reduce a $k$-level projection query to $k$ one-level projection queries, e.g., a $3p$ query with one entity and three relations $e_1 \xrightarrow{r_1} \xrightarrow{r_2} \xrightarrow{r_3} A$ is decomposed to $e_1 \xrightarrow{r_1} A_1, A_1 \xrightarrow{r_2} A_2, A_2 \xrightarrow{r_3} A$.

---

[3] where $k$ is determined by the query type, e.g., for 3-level projection ($3p$) queries, $k = 3$.

- Reduce a $k$-intersection query to $k$ projection queries and an intersection query, e.g., a $3i$ query with intersection of two projection queries $(e_1 \xrightarrow{r_1}) \wedge (e_2 \xrightarrow{r_2}) \wedge (e_3 \xrightarrow{r_3}) = A$ is decomposed to $e_1 \xrightarrow{r_1} A_1, e_2 \xrightarrow{r_2} A_2, e_3 \xrightarrow{r_3} A_2, A_1 \wedge A_2 \wedge A_3 = A$. Similarly, reduce a $k$-union query to $k$ projection queries and a union query.

The complete decomposition of the exhaustive set of query types used in previous work (Ren and Leskovec, 2020) and our empirical studies can be found in Appendix A.

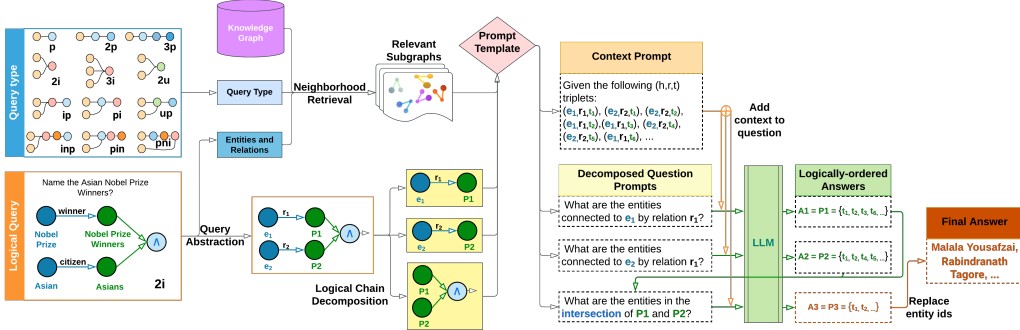

Figure 2: An overview of the LARK model. The model takes the logical query and infers the query type from it. The query abstraction function maps the entities and relations to abstract IDs, and the neighborhood retrieval mechanism collects the relevant subgraphs from the overall knowledge graph. The chains of the abstracted complex query are then logically decomposed to simpler single-operation queries. The retrieved neighborhood and decomposed queries are further converted into LLM prompts using a template and then processed in the LLM to get the final set of answers for evaluation.

### 3.3 Chain Reasoning Prompts

In the previous section, we outlined our approach to limit the neighborhood and decompose complex queries into chains of simple queries. Leveraging these, we can now use the reasoning capability of LLMs to obtain the final set of answers for the query, as shown in Figure 2. To achieve this, we employ a prompt template that converts the neighborhood into a context prompt and the decomposed queries into question prompts. It is worth noting that certain queries in the decomposition depend on the responses of preceding queries, such as intersection relying on the preceding projection queries. Additionally, unlike previous prompting methods such as chain-of-thought (Wei et al., 2022) and decomposition (Khot et al., 2023) prompting, the answers need to be integrated at a certain position in the prompt. To address this issue, we maintain a placeholder in dependent queries and a temporary cache of preceding answers that can replace the placeholders in real-time. This also has the added benefit of maintaining the parallelizability of queries, as we can run batches of decomposed queries in phases instead of sequentially running each decomposed query. The specific prompt templates of the complex and decomposed logical queries for different query types are provided in Appendix B.

### 3.4 Implementation Details

We implemented LARK in Pytorch (Paszke et al., 2019) on eight Nvidia A100 GPUs with 40 GB VRAM. In the case of LLMs, we chose the Llama2 model (Touvron et al., 2023) due to its public availability in the Huggingface library (Wolf et al., 2020) . For efficient inference over the large-scale models, we relied on the mixed-precision version of LLMs and the Deepspeed library (Rasley et al., 2020) with Zero stage 3 optimization. The algorithm of our model is provided in Appendix D and implementation code for all our experiments with exact configuration files and datasets for reproducibility are publicly available[4]. In our experiments, the highest complexity of a query required a 3-hop neighborhood around the entities and relations. Hence, we set the depth limit to 3 (i.e., $k = 3$). Additionally, to further make our process completely compatible with different datasets, we added a limit of $n$ tokens on the input which is dependent on the LLM model (for Llama2, $n$=4096). In practice, this implies that we stop the depth-first traversal when the context becomes longer than $n$.

---

[4] https://anonymous.4open.science/r/LLM-KG-Reasoning-65D1

## 4    Experimental Results

This sections describes our experiments that aim to answer the following research questions (RQs):

**RQ1.** Does LARK outperform the state-of-the-art baselines on the task of logical reasoning over standard knowledge graph benchmarks?
**RQ2.** How does our combination of chain decomposition query and logically-ordered answer mechanism perform in comparison with the standard prompting techniques?
**RQ3.** How does the scale and design of LARK's underlying LLM model affect its performance?
**RQ4.** How would our model perform with support for increased token size?
**RQ5.** Does query abstraction affect the reasoning performance of our model?

### 4.1    Datasets and Baselines

We select the following standard benchmark datasets to investigate the performance of our model against state-of-the-art models on the task of logical reasoning over knowledge graphs:

- **FB15k** (Bollacker et al., 2008) is based on Freebase, a large collaborative knowledge graph project that was created by Google. FB15k contains about 15,000 entities, 1,345 relations, and 592,213 triplets (statements that assert a fact about an entity).
- **FB15k-237** (Toutanova et al., 2015) is a subset of FB15k, containing 14,541 entities, 237 relations, and 310,116 triplets. The relations in FB15k-237 are a subset of the relations in FB15k, and was created to address some of the limitations of FB15k, such as the presence of many irrelevant or ambiguous relations, and to provide a more challenging benchmark for knowledge graph completion models.
- **NELL995** (Carlson et al., 2010) was created using the Never-Ending Language Learning (NELL) system, which is a machine learning system that automatically extracts knowledge from the web by reading text and inferring new facts. NELL995 contains 9,959 entities, 200 relations, and 114,934 triplets. The relations in NELL995 cover a wide range of domains, including geography, sports, and politics.

Our criteria for selecting the above datasets was their ubiquity in previous works on this research problem. Further details on their token size is provided in Appendix E. For the baselines, we chose the following methods:

- **GQE** (Hamilton et al., 2018) encodes a query as a single vector and represents entities and relations in a low-dimensional space. It uses translation and deep set operators, which are modeled as projection and intersection operators, respectively.
- **Query2Box (Q2B)** (Ren et al., 2020) uses a box embedding model which is a generalization of the traditional vector embedding model and can capture richer semantics.
- **BetaE** (Ren and Leskovec, 2020) uses a novel beta distribution to model the uncertainty in the representation of entities and relations. BetaE can capture both the point estimate and the uncertainty of the embeddings, which leads to more accurate predictions in knowledge graph completion tasks.
- **HQE** (Choudhary et al., 2021b) uses the hyperbolic query embedding mechanism to model the complex queries in knowledge graph completion tasks.
- **HypE** (Choudhary et al., 2021b) uses the hyperboloid model to represent entities and relations in a knowledge graph that simultaneously captures their semantic, spatial, and hierarchical features.
- **CQD** (Arakelyan et al., 2021) decomposes complex queries into simpler sub-queries and applies a query-specific attention mechanism to the sub-queries.

### 4.2    RQ1. Efficacy on Logical Reasoning

To study the efficacy of our model on the task of logical reasoning, we compare it against the previous baselines on the following standard logical query constructs:

1. **Multi-hop Projection** traverses multiple relations from a head entity in a knowledge graph to answer complex queries by projecting the query onto the target entities. In our experiments, we consider $1p$, $2p$, and $3p$ queries that denote 1-relation, 2-relation, and 3-relation hop from the head entity, respectively.

2. **Geometric Operations** apply the operations of intersection ($\wedge$) and union ($\vee$) to answer the query. Our experiments use $2i$ and $3i$ queries that represent the intersection over 2 and 3 entities, respectively. Also, we study $2u$ queries that perform union over 2 entities.
3. **Compound Operations** integrate multiple operations such as intersection, union, and projection to handle complex queries over a knowledge graph.
4. **Negation Operations** negate the query by finding entities that do not satisfy the given logic. In our experiments, we examine $2in, 3in, inp,$ and $pin$ queries that negate $2i, 3i, ip,$ and $pi$ queries, respectively. We also analyze $pni$ (an additional variant of the $pi$ query), where the negation is over both entities in the intersection. It should be noted that BetaE is the only method in the existing literature that supports negation, and hence, we only compare against it in our experiments.

We present the results of our experimental study, which compares the Mean Reciprocal Rank (MRR) score of the retrieved candidate entities using different query constructions. MRR is calculated as the average of the reciprocal ranks of the candidate entities [5]. In order to ensure a fair comparison, We selected these query constructions which were used in most of the previous works in this domain (Ren and Leskovec, 2020). An illustration of these query types is provided in Appendix A for better understanding. Our experiments show that LARK outperforms previous state-of-the-art baselines by $35\% - 84\%$ on an average across different query types, as reported in Table 1. We observe that the performance improvement is higher for simpler queries, where $1p > 2p > 3p$ and $2i > 3i$. This suggests that LLMs are better at capturing breadth across relations but may not be as effective at capturing depth over multiple relations. Moreover, our evaluation also encompasses testing against challenging negation queries, for which BetaE (Ren and Leskovec, 2020) remains to be the only existing approach. Even in this complex scenario, our findings, as illustrated in Table 2, indicate that LARK significantly outperforms the baselines by $140\%$. This affirms the superior reasoning capabilities of our model in tackling complex query scenarios. Another point of note is that certain baselines such as CQD are able to outperform LARK in the FB15k dataset for certain query types such as $1p, 3i,$ and $ip$. The reason for this is that FB15k suffers from a data leakage from training to validation and testing sets (Toutanova et al., 2015). This unfairly benefits the training-based baselines over the inference-only LARK model.

Table 1: Performance comparison between LARK and the baseline in terms of their efficacy of logical reasoning using MRR scores. The rows present various models and the columns correspond to different query structures of multi-hop projections, geometric operations, and compound operations. The best results for each query type in every dataset is highlighted in **bold** font.

| Dataset | Models | 1p | 2p | 3p | 2i | 3i | ip | pi | 2u | up |
|---------|--------|------|------|------|------|------|------|------|------|------|
| **FB15k** | GQE | 54.6 | 15.3 | 10.8 | 39.7 | 51.4 | 27.6 | 19.1 | 22.1 | 11.6 |
| | Q2B | 68.0 | 21.0 | 14.2 | 55.1 | 66.5 | 39.4 | 26.1 | 35.1 | 16.7 |
| | BetaE | 65.1 | 25.7 | 24.7 | 55.8 | 66.5 | 43.9 | 28.1 | 40.1 | 25.2 |
| | HQE | 54.3 | 33.9 | 23.3 | 38.4 | 50.6 | 12.5 | 24.9 | 35.0 | 25.9 |
| | HypE | 67.3 | 43.9 | 33.0 | 49.5 | 61.7 | 18.9 | 34.7 | 47.0 | 37.4 |
| | CQD | **79.4** | 39.6 | 27.0 | **74.0** | **78.2** | **70.0** | 43.3 | 48.4 | 17.5 |
| | LARK(complex) | 73.6 | 46.5 | 32.0 | 66.9 | 61.8 | 24.8 | 47.2 | 47.7 | 37.5 |
| | LARK(ours) | 73.6 | **49.3** | **35.1** | 67.8 | 62.6 | 29.3 | **54.5** | **51.9** | **37.7** |
| **FB15k-237** | GQE | 35.0 | 7.2 | 5.3 | 23.3 | 34.6 | 16.5 | 10.7 | 8.2 | 5.7 |
| | Q2B | 40.6 | 9.4 | 6.8 | 29.5 | 42.3 | 21.2 | 12.6 | 11.3 | 7.6 |
| | BetaE | 39.0 | 10.9 | 10.0 | 28.8 | 42.5 | 22.4 | 12.6 | 12.4 | 9.7 |
| | HQE | 37.6 | 20.9 | 16.9 | 25.3 | 35.2 | 17.3 | 8.2 | 15.6 | 17.9 |
| | HypE | 49.0 | 34.3 | 23.7 | 33.9 | 44 | 18.6 | 30.5 | 41.0 | 26.0 |
| | CQD | 44.5 | 11.3 | 8.1 | 32.0 | 42.7 | 25.3 | 15.3 | 13.4 | 4.8 |
| | LARK(complex) | **70.0** | 34.0 | 21.5 | 43.4 | 42.2 | 18.7 | 38.4 | 49.2 | 25.1 |
| | LARK(ours) | **70.0** | **36.9** | **24.5** | **44.3** | **43.1** | **23.2** | **45.6** | **56.6** | **25.4** |
| **NELL995** | GQE | 32.8 | 11.9 | 9.6 | 27.5 | 35.2 | 18.4 | 14.4 | 8.5 | 8.8 |
| | Q2B | 42.2 | 14.0 | 11.2 | 33.3 | 44.5 | 22.4 | 16.8 | 11.3 | 10.3 |
| | BetaE | 53.0 | 13.0 | 11.4 | 37.6 | 47.5 | 24.1 | 14.3 | 12.2 | 8.5 |
| | HQE | 35.5 | 20.9 | 18.9 | 23.2 | 36.3 | 8.8 | 13.7 | 21.3 | 15.5 |
| | HypE | 46.0 | 30.6 | 27.9 | 33.6 | 48.6 | 31.8 | 13.5 | 20.7 | 26.4 |
| | CQD | 50.7 | 18.4 | 13.8 | 39.8 | **49.0** | **29.0** | 22.0 | 16.3 | 9.9 |
| | LARK(complex) | **83.2** | 39.8 | 27.6 | 49.3 | 48.0 | 18.7 | 19.6 | 8.3 | 36.8 |
| | LARK(ours) | **83.2** | **42.3** | **31.0** | **49.9** | 48.7 | 23.1 | **23.0** | 20.1 | 37.2 |

---

[5]More metrics such as HITS@K=1,3,10 are reported in Appendix C.

Table 2: Performance comparison between LARK and the baseline for negation query types using MRR scores. The best results for each query type in every dataset is highlighted in **bold** font. Our model's performance is significantly higher on most negation queries. However, the performance is limited in *3in* and *pni* queries due to their high number of tokens (shown in Appendix E).

| Dataset | Models | 2in | 3in | inp | pin | pni |
|---|---|---|---|---|---|---|
| **FB15k** | **BetaE** | 14.3 | **14.7** | 11.5 | 6.5 | **12.4** |
| | **LARK(complex)** | 16.5 | 6.2 | 32.5 | 22.8 | 10.5 |
| | **LARK(ours)** | **17.5** | 7.0 | **34.7** | **26.7** | 11.1 |
| **FB15k-237** | **BetaE** | 5.1 | **7.9** | 7.4 | 3.6 | 3.4 |
| | **LARK(complex)** | 6.1 | 3.4 | 21.6 | 12.8 | 2.9 |
| | **LARK(ours)** | **7.0** | 4.1 | **23.9** | **16.8** | **3.5** |
| **NELL995** | **BetaE** | 5.1 | **7.8** | 10.0 | 3.1 | 3.5 |
| | **LARK(complex)** | 8.9 | 5.3 | 23.0 | 10.4 | 6.3 |
| | **LARK(ours)** | **10.4** | 6.6 | **25.4** | **13.6** | **7.6** |

## 4.3 RQ2. ADVANTAGES OF CHAIN DECOMPOSITION

The aim of this experiment is to investigate the advantages of using chain decomposed queries over standard complex queries. We employ the same experimental setup described in Section 4.2. Our results, in Tables 1 and 2, demonstrate that utilizing chain decomposition contributes to a significant improvement of $20\% - 33\%$ in our model's performance. This improvement is a clear indication of the LLMs' ability to capture a broad range of relations and effectively utilize this capability for enhancing the performance on complex queries. This study highlights the potential of using chain decomposition to overcome the limitations of complex queries and improve the efficiency of logical reasoning tasks. This finding is a significant contribution to the field of natural language processing and has implications for various other applications such as question-answering systems and knowledge graph completion. Overall, our results suggest that chain-decomposed queries could be a promising approach for improving the performance of LLMs on complex logical reasoning tasks.

## 4.4 RQ3. ANALYSIS OF LLM SCALE

This experiment analyzes the impact of the size of the underlying LLMs and query abstraction on the overall LARK model performance. To examine the effect of LLM size, we compared two variants of the Llama2 model which have 7 billion and 13 billion parameters. Our evaluation results, presented in Table 3, show that the performance of the LARK model improves by $123\%$ from Llama2-7B to Llama2-13B. This indicates that increasing the number of LLM parameters can enhance the performance of LARK model.

Table 3: MRR scores of LARK on FB15k-237 dataset with underlying LLMs of different sizes. The best results for each query type is highlighted in **bold** font.

| LLM | # Params | 1p | 2p | 3p | 2i | 3i | ip | pi | 2u | up | 2in | 3in | inp | pin | pni |
|---|---|---|---|---|---|---|---|---|---|---|---|---|---|---|---|
| **Llama2** | 7B | 73.1 | 33.2 | 20.6 | 10.6 | 25.2 | 25.9 | 17.2 | 20.8 | 24.3 | 4 | 1.8 | 14.2 | 7.4 | 1.9 |
| | 13B | **73.6** | **49.3** | **35.1** | **67.8** | **62.6** | **29.3** | **54.5** | **51.9** | **37.7** | **7.0** | **4.1** | **23.9** | **16.8** | **3.5** |

## 4.5 RQ4. STUDY ON INCREASED TOKEN LIMIT OF LLMS

From the dataset details provided in Appendix E, we observe that the token size of different query types shows considerable fluctuation from $58$ to over $100,000$. Unfortunately, the token limit of LLama2, considered as the base in our experiments, is 4096. This limit is insufficient to demonstrate the full potential performance of LARK on our tasks. To address this limitation, we consider the availability of models with higher token limits, such as GPT-3.5 (OpenAI, 2023). However, we acknowledge that these models are expensive to run and thus, we could not conduct a thorough analysis on the entire dataset. Nevertheless, to gain insight into LARK's potential with increased token size, we randomly sampled 1000 queries per query type from each dataset with token length over 4096 and less than 4096 and compared our model on these queries with GPT-3.5 and Llama2 as the base. The evaluation results, which are displayed in Table 4, demonstrate that transitioning from Llama2 to GPT-3.5 can lead to a significant performance improvement of 29%-40% for the LARK model which suggests that increasing the token limit of LLMs may have significant potential of further performance enhancement.

Table 4: MRR scores of LARK with Llama2 and GPT LLMs as the underlying base models. The best results for each query type in every dataset is highlighted in **bold** font.

| | FB15k | | | | | | | | | | | | | |
|---|---|---|---|---|---|---|---|---|---|---|---|---|---|---|
| **LLM** | **1p** | **2p** | **3p** | **2i** | **3i** | **ip** | **pi** | **2u** | **up** | **2in** | **3in** | **inp** | **pin** | **pni** |
| **Llama2-7B** | 23.4 | 21.5 | 22.6 | 3.4 | 3 | 26.1 | 18.4 | 14.8 | 3.9 | 9.5 | 4.7 | 21.7 | 26.4 | 5.8 |
| **Llama2-13B** | 23.8 | 22.8 | 24.2 | 3.5 | 3 | 23.3 | 30.8 | 30.7 | 3.9 | 12.4 | 6.6 | 28.4 | 51.4 | 7.7 |
| **GPT-3.5** | **36.1** | **34.6** | **36.8** | **17.0** | **14.4** | **35.4** | **46.7** | **39.3** | **19.5** | **18.8** | **10.0** | **43.1** | **56.7** | **11.6** |
| | FB15k-237 | | | | | | | | | | | | | |
| **LLM** | **1p** | **2p** | **3p** | **2i** | **3i** | **ip** | **pi** | **2u** | **up** | **2in** | **3in** | **inp** | **pin** | **pni** |
| **Llama2-7B** | 23.1 | 27.4 | 31.5 | 5 | 4.1 | 26.6 | 20.9 | 15.3 | 5.6 | 26.6 | 8.8 | 33.7 | 31 | 21.1 |
| **Llama2-13B** | 23.5 | 29.2 | 33.8 | 5 | 4.1 | 23.7 | 35 | 31.7 | 5.6 | 34.7 | 12.3 | 44 | 60.4 | 28 |
| **GPT-3.5** | **35.7** | **44.2** | **51.2** | **24.8** | **20.2** | **36.0** | **53.1** | **40.6** | **28.1** | **52.5** | **18.7** | **66.8** | **66.6** | **42.4** |
| | NELL995 | | | | | | | | | | | | | |
| **LLM** | **1p** | **2p** | **3p** | **2i** | **3i** | **ip** | **pi** | **2u** | **up** | **2in** | **3in** | **inp** | **pin** | **pni** |
| **Llama2-7B** | 28 | 24.4 | 27.6 | 3.7 | 3.2 | 24 | 8.4 | 14.5 | 5.7 | 14 | 7.7 | 23.1 | 21.3 | 13.4 |
| **Llama2-13B** | 28.4 | 26 | 29.5 | 3.7 | 3.2 | 21.5 | 14.1 | 25.4 | 5.7 | 18.3 | 10.8 | 30.1 | 30.2 | 17.7 |
| **GPT-3.5** | **43.1** | **39.4** | **44.8** | **18.3** | **15.5** | **32.6** | **21.4** | **38.5** | **28.3** | **27.7** | **16.4** | **45.7** | **45.9** | **26.8** |

## 4.6 RQ5. EFFECTS OF QUERY ABSTRACTION

Regarding the analysis of query abstraction, we considered a variant of LARK called LARK (semantic), which retains semantic information in KG entities and relations. As shown in Figure 3, we observe that semantic information provides a minor performance enhancement of 0.01% for simple projection queries. However, in more complex queries, it results in a performance degradation of 0.7% − 1.4%. The primary cause of this degradation is that the inclusion of semantic information exceeds the LLMs' token limit, leading to a loss of neighborhood information. Hence, we assert that query abstraction is not only a valuable technique for mitigating model hallucination and achieving generalization across different KG datasets but can also enhance performance by reducing token size.

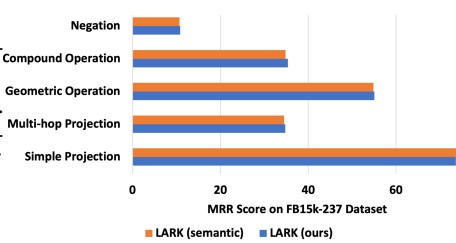

Figure 3: Effects of Query Abstraction.

## 5 CONCLUDING DISCUSSION

In this paper, we presented LARK, the first approach to integrate logical reasoning over knowledge graphs with the capabilities of LLMs. Our approach utilizes logically-decomposed LLM prompts to enable chain reasoning over subgraphs retrieved from knowledge graphs, allowing us to efficiently leverage the reasoning ability of LLMs. Through our experiments on logical reasoning across standard KG datasets, we demonstrated that LARK outperforms previous state-of-the-art approaches by a significant margin on 14 different FOL query types. Finally, our work also showed that the performance of LARK improves with increasing scale and better design of the underlying LLMs. We demonstrated that LLMs that can handle larger input token lengths can lead to significant performance improvements. Overall, our approach presents a promising direction for integrating LLMs with logical reasoning over knowledge graphs.

The proposed approach of using Large Language Models (LLMs) for complex logical reasoning over Knowledge Graphs (KGs) is expected to pave a new way for improved reasoning over large, noisy, and incomplete real-world KGs. This can potentially have a significant impact on various applications such as natural language understanding, question answering systems, and intelligent information retrieval systems, etc. For example, in healthcare, KGs can be used to represent patient data, medical knowledge, and clinical research, and logical reasoning over these KGs can enable better diagnosis, treatment, and drug discovery. However, there are also ethical considerations to be taken into account. As with most AI-based technology, there is a potential risk of inducing bias into the model, which can lead to unfair decisions and actions. Bias can be introduced in the KGs themselves, as they are often created semi-automatically from biased sources, and can be amplified by the logical reasoning process. Moreover, the large amount of data used to train LLMs can also introduce bias, as it may reflect societal prejudices and stereotypes. Therefore, it is essential to carefully monitor and evaluate the KGs and LLMs used in this approach to ensure fairness and avoid discrimination. The performance of this method is also dependent on the quality and completeness of the KGs used, and the limited token size of current LLMs. But, we also observe that the current trend of increasing LLM token limits will soon resolve some of these limitations.

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
