# APPENDIX

## A    QUERY DECOMPOSITION OF DIFFERENT QUERY TYPES

Figure 4 provides the query decomposition of different query types considered in our empirical study as well as previous literature in the area.

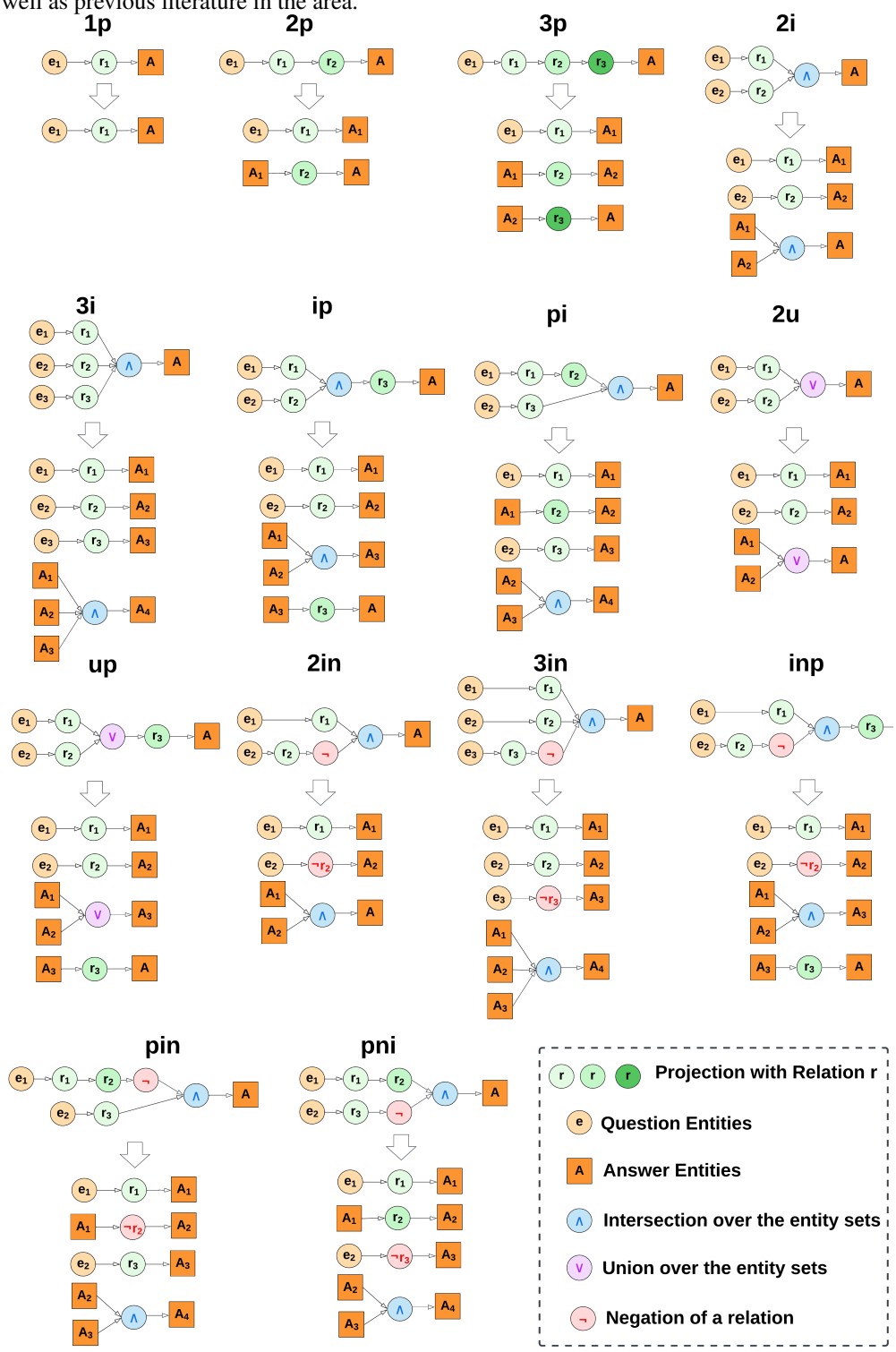

Figure 4: Query Decomposition of different query types considered in our experiments.

## B  PROMPT TEMPLATES OF DIFFERENT QUERY TYPES

The prompt templates for full complex logical queries with multiple operations and decomposed elementary logical queries with single operation are provided in Tables 5 and 6, respectively.

Table 5: Full Prompt Templates of Different Query Types.

| Type | Logical Query | Template for Full Prompts |
|---|---|---|
| Context | $\mathcal{N}_k(q_\tau[Q_\tau])$ | Given the following (h,r,t) triplets where entity h is related to entity t by relation r; $(h_1, r_1, t_1), (h_2, r_2, t_2), (h_3, r_3, t_3), (h_4, r_4, t_4),$ $(h_5, r_5, t_5), (h_6, r_6, t_6), (h_7, r_7, t_7), (h_8, r_8, t_8)$ |
| 1p | $\exists X.r_1(X, e_1)$ | Which entities are connected to $e_1$ by relation $r_1$? |
| 2p | $\exists X.r_1(X, \exists Y.r_2(Y, e_1)$ | Let us assume that the set of entities E is connected to entity $e_1$ by relation $r_1$. Then, what are the entities connected to E by relation $r_2$? |
| 3p | $\exists X.r_1(X, \exists Y.r_2(Y, \exists Z.r_3(Z, e_1)$ | Let us assume that the set of entities E is connected to entity $e_1$ by relation $r_1$ and the set of entities F is connected to entities in E by relation $r_2$. Then, what are the entities connected to F by relation $r_3$? |
| 2i | $\exists X.[r_1(X, e_1) \wedge r_2(X, e_2)]$ | Let us assume that the set of entities E is connected to entity $e_1$ by relation $r_1$ and the set of entities F is connected to entity $e_2$ by relation $r_2$. Then, what are the entities in the intersection of set E and F, i.e., entities present in both F and G? |
| 3i | $\exists X.[r_1(X, e_1) \wedge r_2(X, e_2) \wedge r_3(X, e_3)]$ | Let us assume that the set of entities E is connected to entity $e_1$ by relation $r_1$, the set of entities F is connected to entity $e_2$ by relation $r_2$ and the set of entities G is connected to entity $e_3$ by relation $r_3$. Then, what are the entities in the intersection of set E, F and G, i.e., entities present in all E, F and G? |
| ip | $\exists X.r_3(X, \exists Y.[r_1(Y, e_1) \wedge r_2(Y, e_2)]$ | Let us assume that the set of entities E is connected to entity $e_1$ by relation $r_1$, F is the set of entities connected to entity $e_2$ by relation $r_2$, and G is the set of entities in the intersection of E and F. Then, what are the entities connected to entities in set G by relation $r_3$? |
| pi | $\exists X.[r_1(X, \exists Y.r_2(Y, e_2)) \wedge r_3(X, e_3)]$ | Let us assume that the set of entities E is connected to entity $e_1$ by relation $r_1$, F is the set of entities connected to entities in E by relation $r_2$, and G is the set of entities connected to entity $e_2$ by relation $r_3$. Then, what are the entities in the intersection of set F and G, i.e., entities present in both F and G? |
| 2u | $\exists X.[r_1(X, e_1) \vee r_2(X, e_2)]$ | Let us assume that the set of entities E is connected to entity $e_1$ by relation $r_1$ and F is the set of entities connected to entity $e_2$ by relation $r_2$. Then, what are the entities in the union of set F and G, i.e., entities present in either F or G? |
| up | $\exists X.r_3(X, \exists Y.[r_1(Y, e_1) \vee r_2(Y, e_2)]$ | Let us assume that the set of entities E is connected to entity $e_1$ by relation $r_1$ and F is the set of entities connected to entity $e_2$ by relation $r_2$. G is the set of entities in the union of E and F. Then, what are the entities connected to entities in G by relation $r_3$? |
| 2in | $\exists X.[r_1(X, e_1) \wedge \neg r_2(X, e_2)]$ | Let us assume that the set of entities E is connected to entity $e_1$ by relation $r_1$ and F is the set of entities connected to entity $e_2$ by any relation other than relation $r_2$. Then, what are the entities in the intersection of set E and F, i.e., entities present in both F and G? |
| 3in | $\exists X.[r_1(X, e_1) \wedge r_2(X, e_2) \wedge \neg r_3(X, e_3)]$ | Let us assume that the set of entities E is connected to entity $e_1$ by relation $r_1$, F is the set of entities connected to entity $e_2$ by relation $r_2$, and F is the set of entities connected to entity $e_3$ by any relation other than relation $r_3$. Then, what are the entities in the intersection of set E and F, i.e., entities present in both F and G? |
| inp | $\exists X.r_3(X, \exists Y.[r_1(Y, e_1) \wedge \neg r_2(Y, e_2)]$ | Let us assume that the set of entities E is connected to entity $e_1$ by relation $r_1$, and F is the set of entities connected to entity $e_2$ by any relation other than relation $r_2$. Then, what are the entities that are connected to the entities in the intersection of set E and F by relation $r_3$? |
| pin | $\exists X.[r_1(X, \exists Y.\neg r_2(Y, e_2)) \wedge r_3(X, e_3)]$ | Let us assume that the set of entities E is connected to entity $e_1$ by relation $r_1$, F is the set of entities connected to entities in E by relation $r_2$, and G is the set of entities connected to entity $e_2$ by any relation other than relation $r_3$. Then, what are the entities in the intersection of set F and G, i.e., entities present in both F and G? |
| pni | $\exists X.[r_1(X, \exists Y.\neg r_2(Y, e_2)) \wedge \neg r_3(X, e_3)]$ | Let us assume that the set of entities E is connected to entity $e_1$ by relation $r_1$, F is the set of entities connected to entities in E by any relation other than $r_2$, and G is the set of entities connected to entity $e_2$ by relation $r_3$. Then, what are the entities in the intersection of set F and G, i.e., entities present in both F and G? |

## C  ANALYSIS OF LOGICAL REASONING PERFORMANCE USING HITS METRIC

Tables 7 and 8 present the HITS@K=3 results of baselines and our model. HITS@K indicates the accuracy of predicting correct candidates in the top-K results.

Table 6: Decomposed Prompt Templates of Different Query Types.

| Type | Logical Query | Template for Decomposed Prompts |
|---|---|---|
| **Context** | $\mathcal{N}_k(q_\tau[Q_\tau])$ | Given the following (h,r,t) triplets where entity h is related to entity t by relation r; $(h_1, r_1, t_1), (h_2, r_2, t_2), (h_3, r_3, t_3), (h_4, r_4, t_4),$ $(h_5, r_5, t_5), (h_6, r_6, t_6), (h_7, r_7, t_7), (h_8, r_8, t_8)$ |
| **1p** | $\exists X.r_1(X, e_1)$ | Which entities are connected to $e_1$ by relation $r_1$? |
| **2p** | $\exists X.r_1(X, \exists Y.$ $r_2(Y, e_1)$ | Which entities are connected to $e_1$ by relation $r_1$? Which entities are connected to any entity in [PP1] by relation $r_2$? |
| **3p** | $\exists X.r_1(X, \exists Y$ $.r_2(Y, \exists Z.$ $r_3(Z, e_1)$ | Which entities are connected to $e_1$ by relation $r_1$? Which entities are connected to any entity in [PP1] by relation $r_2$? Which entities are connected to any entity in [PP2] by relation $r_3$? |
| **2i** | $\exists X.[r_1(X, e_1)$ $\wedge r_2(X, e_2)]$ | Which entities are connected to $e_1$ by relation $r_1$? Which entities are connected to $e_2$ by relation $r_2$? What are the entities in the intersection of entity sets [PP1] and [PP2]? |
| **3i** | $\exists X.[r_1(X, e_1)$ $\wedge r_2(X, e_2)$ $\wedge r_3(X, e_3)]$ | Which entities are connected to $e_1$ by relation $r_1$? Which entities are connected to $e_2$ by relation $r_2$? Which entities are connected to $e_3$ by relation $r_3$? What are the entities in the intersection of entity sets [PP1], [PP2] and [PP3]? |
| **ip** | $\exists X.r_3(X, \exists Y.[r_1(Y, e_1)$ $\wedge r_2(Y, e_2)]$ | Which entities are connected to $e_1$ by relation $r_1$? Which entities are connected to $e_2$ by relation $r_2$? What are the entities in the intersection of entity sets [PP1] and [PP2]? What are the entities connected to any entity in [PP3] by relation $r_3$? |
| **pi** | $\exists X.[r_1(X, \exists Y.r_2(Y, e_2))$ $\wedge r_3(X, e_3)]$ | Which entities are connected to $e_1$ by relation $r_1$? Which entities are connected to [PP1] by relation $r_2$? Which entities are connected to $e_2$ by relation $r_3$? What are the entities in the intersection of entity sets [PP2] and [PP3]? |
| **2u** | $\exists X.[r_1(X, e_1)$ $\vee r_2(X, e_2)]$ | Which entities are connected to $e_1$ by relation $r_1$? Which entities are connected to $e_2$ by relation $r_2$? What are the entities in the union of entity sets [PP1] and [PP2]? |
| **up** | $\exists X.r_3(X, \exists Y.[r_1(Y, e_1)$ $\vee r_2(Y, e_2)]$ | Which entities are connected to $e_1$ by relation $r_1$? Which entities are connected to $e_2$ by relation $r_2$? What are the entities in the union of entity sets [PP1] and [PP2]? Which entities are connected to any entity in [PP3] by relation $r_3$? |
| **2in** | $\exists X.[r_1(X, e_1)$ $\wedge \neg r_2(X, e_2)]$ | Which entities are connected to $e_1$ by any relation other than $r_1$? Which entities are connected to $e_2$ by any relation other than $r_2$? What are the entities in the intersection of entity sets [PP1] and [PP2]? |
| **3in** | $\exists X.[r_1(X, e_1)$ $\wedge r_2(X, e_2)$ $\wedge \neg r_3(X, e_3)]$ | Which entities are connected to $e_1$ by any relation other than $r_1$? Which entities are connected to $e_2$ by any relation other than $r_2$? Which entities are connected to $e_3$ by any relation other than $r_3$? What are the entities in the intersection of entity sets [PP1], [PP2] and [PP3]? |
| **inp** | $\exists X.r_3(X, \exists Y.[r_1(Y, e_1)$ $\wedge \neg r_2(Y, e_2)]$ | Which entities are connected to $e_1$ by relation $r_1$? Which entities are connected to $e_2$ by any relation other than $r_2$? What are the entities in the intersection of entity sets [PP1], and [PP2]? What are the entities connected to any entity in [PP3] by relation $r_3$? |
| **pin** | $\exists X.[r_1(X, \exists Y.\neg r_2(Y, e_2))$ $\wedge r_3(X, e_3)]$ | Which entities are connected to $e_1$ by relation $r_1$? Which entities are connected to entity set in [PP1] by relation $r_2$? Which entities are connected to $e_2$ by any relation other than $r_3$? What are the entities in the intersection of entity sets [PP2] and [PP3]? |
| **pni** | $\exists X.[r_1(X, \exists Y.\neg r_2(Y, e_2))$ $\wedge \neg r_3(X, e_3)]$ | Which entities are connected to $e_1$ by relation $r_1$? Which entities are connected to any entity in [PP1] by any relation other than $r_2$? Which entities are connected to $e_2$ by relation $r_3$? What are the entities in the intersection of entity sets [PP2] and [PP3]? |

Table 7: Performance comparison study between LARK and the baseline, focusing on their efficacy of logical reasoning using HITS@K=1,3,10 scores. The rows correspond to the models and columns denote the different query structures of multi-hop projections, geometric operations, and compound operations. The best results for each query type in every dataset are highlighted in **bold** font.

| Dataset | Variant | 1p | 2p | 3p | 2i | 3i | ip | pi | 2u | up |
|---|---|---|---|---|---|---|---|---|---|---|
| | | | | | | HITS@1 | | | | |
| FB15k | Llama2-7B | 74.6 | 26 | 18.5 | 59.9 | 47.7 | 2.4 | 5.7 | 5.8 | 5 |
| | complex | **77.5** | 37.9 | 26.3 | 67.4 | 54.6 | 8.2 | 20.7 | 20.7 | 17.6 |
| | step | **77.5** | **41.8** | **28.1** | **70.2** | **57.3** | **10.3** | **24.3** | **22.8** | **17.8** |
| FB15k-237 | Llama2-7B | 77.2 | 28.5 | 17.7 | 10.9 | 22.6 | 10.8 | 8.7 | 10.5 | 13.2 |
| | complex | **78.5** | 30.8 | 19.3 | 41.1 | 38.1 | 9.6 | 18.7 | 24.2 | 14.0 |
| | step | **78.5** | **34.3** | **21.3** | **43.2** | **40.2** | **11.7** | **22.2** | **27.9** | **14.2** |
| NELL995 | Llama2-7B | 86.4 | 28.3 | 19.6 | 10.2 | 24 | 8.6 | 3.5 | 1.5 | 15.9 |
| | complex | **88.0** | 30.9 | 21.7 | 44.1 | 41.6 | 7.4 | 8.2 | 3.3 | 17 |
| | step | **88.0** | **34.3** | **24.0** | **46.1** | **43.8** | **9.5** | **9.8** | **8.9** | **17.3** |
| | | | | | | HITS@3 | | | | |
| FB15k | Llama2-7B | 74 | 53.4 | 34.6 | 18.2 | 36.4 | 44.7 | 39.4 | 35.7 | 77.1 |
| | complex | **77.7** | **57.6** | 37.9 | 68.5 | 61.3 | 39.6 | 84.8 | 82.9 | 81.7 |
| | step | **77.7** | 57.4 | **40.1** | **69.4** | **62.5** | **48.4** | **91.2** | **92.7** | **82.6** |
| FB15k-237 | Llama2-7B | 75.9 | 42.6 | 25.7 | 12.6 | 25.9 | 43.6 | 35.1 | 42.9 | 53.8 |
| | complex | **78.3** | **45.9** | 28.1 | 47.2 | 43.7 | 38.7 | 75.6 | 89.4 | 57 |
| | step | **78.3** | **45.9** | **29.8** | **48.2** | **44.6** | **47.3** | **80.0** | **93.6** | **57.6** |
| NELL995 | Llama2-7B | 85.6 | 42.9 | 28.7 | 11.8 | 27.6 | 34.6 | 14.1 | 5.7 | 65 |
| | complex | **87.8** | **46.8** | 31.6 | 50.7 | 47.9 | 29.8 | 32.9 | 13.2 | 69.4 |
| | step | **87.8** | 45.7 | **33.5** | **51.3** | **48.7** | **38.1** | **39.6** | **35.8** | **70.3** |
| | | | | | | HITS@10 | | | | |
| FB15k | Llama2-7B | 73.6 | 53.9 | 35.7 | 18.1 | 36.3 | 44.6 | 39.5 | 35.7 | 77.1 |
| | complex | **77.7** | **58.2** | 39.1 | 68.2 | 61.4 | 39.5 | 85 | 82.9 | 81.7 |
| | step | **77.7** | 57.4 | **46.0** | **69.4** | **62.5** | **48.2** | **91.2** | **84.7** | **82.6** |
| FB15k-237 | Llama2-7B | 75.2 | 43 | 26.5 | 12.6 | 25.9 | 43.6 | 35.1 | 42.9 | 53.8 |
| | complex | **78.3** | **46.4** | 29 | 47.3 | 43.8 | 38.7 | 75.6 | 89.4 | 57 |
| | step | **78.3** | 45.9 | **34.1** | **48.2** | **44.6** | **47.3** | **80.0** | **93.6** | **57.6** |
| NELL995 | Llama2-7B | 84.9 | 43.4 | 29.2 | 11.8 | 27.6 | 34.6 | 14.1 | 5.7 | 65 |
| | complex | **87.8** | **47.4** | 32.2 | 50.8 | 48 | 29.8 | 32.9 | 13.2 | 69.4 |
| | step | **87.8** | 45.7 | **38.3** | **51.3** | **48.7** | **38.1** | **39.6** | **35.8** | **70.3** |

Table 8: Performance comparison between LARK and the baseline for negation query types using HITS@K=1,3,10 scores. The best results for each query type in every dataset are given in **bold** font.

| Metric | Variant | 2in | 3in | inp | pin | pni | 2in | 3in | inp | pin | pni | 2in | 3in | inp | pin | pni |
|---|---|---|---|---|---|---|---|---|---|---|---|---|---|---|---|---|
| | | | | HITS@1 | | | | | HITS@3 | | | | | HITS@10 | | |
| FB15k | Llama2-7B | 1.8 | 0.7 | 4.0 | 2.1 | 0.9 | 18.6 | 5.7 | 40.8 | 18.8 | 8.6 | 18.6 | 5.7 | 40.8 | 18.8 | 8.6 |
| | complex | 6.7 | 2.4 | 14.2 | 7.8 | 3.3 | 26.6 | 9.5 | 59.2 | 30.3 | 12.3 | 26.6 | 9.5 | 59.3 | 30.3 | 12.4 |
| | step | **7.4** | **2.7** | **14.9** | **9.1** | **3.4** | **31.0** | **12.1** | **64.8** | **38.7** | **14.4** | **31.0** | **12.1** | **64.8** | **38.7** | **14.4** |
| FB15k-237 | Llama2-7B | 1.9 | 0.8 | 6.8 | 2.8 | 0.7 | 7.5 | 3.5 | 27.3 | 11.6 | 2.7 | 7.5 | 3.5 | 27.3 | 11.6 | 2.7 |
| | complex | 2.7 | 1.4 | 9.8 | 4.6 | 1 | 10.8 | 5.8 | 39.6 | 18.7 | 3.9 | 10.8 | 5.8 | 39.6 | 18.7 | 3.9 |
| | step | **3.2** | **1.7** | **10.6** | **5.8** | **1.1** | **12.6** | **7.4** | **43.3** | **23.9** | **4.6** | **12.6** | **7.4** | **43.3** | **23.9** | **4.6** |
| NELL995 | Llama2-7B | 2.8 | 1.4 | 7.2 | 2.2 | 1.5 | 11.2 | 6 | 29.1 | 9.2 | 6.2 | 11.2 | 6 | 29.1 | 9.2 | 6.2 |
| | complex | 3.9 | 2.3 | 10.2 | 3.7 | 2.2 | 16.1 | 9.4 | 41.8 | 15.1 | 9 | 16.1 | 9.4 | 41.8 | 15.1 | 9 |
| | step | **4.6** | **2.8** | **11.1** | **4.7** | **2.7** | **18.5** | **12.0** | **46.0** | **19.3** | **10.9** | **18.5** | **12.0** | **46.0** | **19.3** | **10.9** |

## D  ALGORITHM

Algorithm for the LARK's procedure is provided in Algorithm 1.

---
**Algorithm 1:** LARK Algorithm

---
**Input:** Logical query $q_\tau$, Knowledge Graph $\mathcal{G} : E \times R$;
**Output:** Answer entities $V_\tau$;
1 # Query Abstraction: Map entity and relations to IDs
2 $q_\tau = Abstract(q_\tau)$;
3 $\mathcal{G} = Abstract(\mathcal{G})$;
4 # Neighborhood Retrieval
5 $\mathcal{N}_k(q_\tau[Q_\tau]) = \{(h, r, t)\}$, using Eq. (7)
6 # Query Decomposition
7 $q_\tau^d = Decomp(q_\tau)$;
8 # Initialize Answer Cache $ans = \{\}$;
9 **for** $i \in 1 : length\left(q_\tau^d\right)$ **do**
10    # Replace Answer Cache in Question
11    $q_\tau^d[i] = replace(q_\tau^d[i], ans[i-1])$;
12    $ans[i] = LLM\left(q_\tau^d[i]\right)$;
13 **end**
14 **return** $ans[length\left(q_\tau^d\right)]$

---

Table 9: Details of the token distribution for various query types in different datasets. The columns present the mean, median, minimum (Min), and maximum (Max) values of the number of tokens in the queries of different query types. Column 'Cov' presents the percentage of queries (coverage) that contain less than 4096 tokens, which is the token limit of Llama2 model.

| Dataset | FB15k | | | | | FB15k-237 | | | | | NELL | | | | |
|---|---|---|---|---|---|---|---|---|---|---|---|---|---|---|---|
| Type | Mean | Median | Min | Max | Cov | Mean | Median | Min | Max | Cov | Mean | Median | Min | Max | Cov |
| 1p | 70.2 | 61 | 58 | 10338 | 100 | 82.1 | 61 | 58 | 30326 | 99.9 | 81.7 | 61 | 58 | 30250 | 99.9 |
| 2p | 331.2 | 106 | 86 | 27549 | 97.1 | 1420.9 | 140 | 83 | 130044 | 89.7 | 893.4 | 136 | 83 | 108950 | 90.9 |
| 3p | 785.2 | 165 | 103 | 80665 | 91 | 3579.8 | 329 | 103 | 208616 | 75.7 | 3052.6 | 389 | 100 | 164545 | 73.7 |
| 2i | 1136.7 | 276 | 119 | 20039 | 86.3 | 4482.8 | 636 | 119 | 60655 | 67.7 | 4469.3 | 680 | 119 | 54916 | 67.3 |
| 3i | 2575.4 | 860 | 145 | 29148 | 68.4 | 8760.2 | 2294 | 145 | 85326 | 48.3 | 8979.4 | 2856 | 145 | 76834 | 44.8 |
| ip | 1923.8 | 1235 | 135 | 21048 | 67.4 | 4035.8 | 2017 | 131 | 32795 | 50.5 | 4838 | 2676 | 131 | 33271 | 43.6 |
| pi | 1036.8 | 455 | 140 | 10937 | 85.8 | 1255.6 | 343 | 141 | 45769 | 83.4 | 1535.3 | 435 | 135 | 21125 | 79.9 |
| 2u | 1325.4 | 790 | 121 | 14703 | 80.8 | 2109.5 | 868 | 123 | 60655 | 68.9 | 2294.9 | 1138 | 125 | 23637 | 65.7 |
| up | 115.3 | 112 | 110 | 958 | 100 | 113.7 | 112 | 110 | 981 | 100 | 113.2 | 112 | 110 | 427 | 100 |
| 2in | 1169.1 | 548 | 123 | 18016 | 84.9 | 5264.7 | 1116 | 128 | 60281 | 61.8 | 3496 | 774 | 124 | 58032 | 71.6 |
| 3in | 4070.3 | 2230 | 159 | 28679 | 46.6 | 13695.8 | 8344 | 175 | 88561 | 25.9 | 12575.9 | 7061 | 164 | 88250 | 28.1 |
| inp | 629 | 112 | 110 | 73457 | 91.8 | 1949.4 | 394 | 110 | 115169 | 78.4 | 696.7 | 112 | 110 | 89660 | 93.8 |
| pin | 400.7 | 154 | 129 | 6802 | 95.8 | 1106.5 | 242 | 129 | 44010 | 87.2 | 418.1 | 131 | 129 | 24062 | 96.7 |
| pni | 345.9 | 129 | 127 | 7938 | 96.6 | 547.1 | 129 | 127 | 18057 | 95.1 | 289.3 | 129 | 127 | 17489 | 97.9 |

## E  QUERY TOKEN DISTRIBUTION IN DATASETS

The quantitative details of the query token's lengths is provided in Table 9 and their complete distribution plots are provided in Figure 5. From the results, we observe that the distribution of token lengths is positively-skewed for most of the query types, which indicates that the number of samples with high token lengths is small in number. Thus, small improvements in the LLMs' token limit can potentially lead to better coverage on most of the reasoning queries in standard KG datasets.

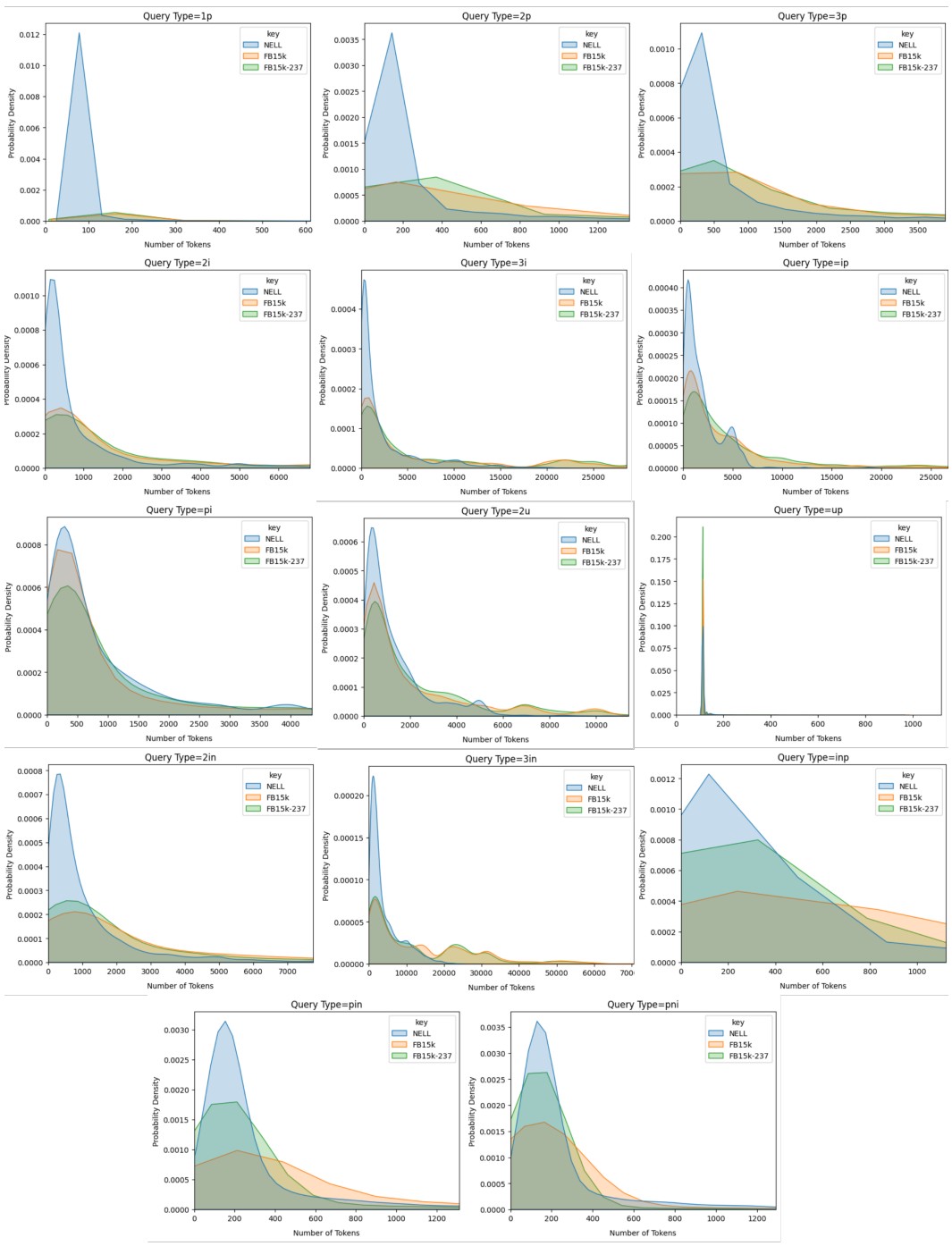

Figure 5: Probability distribution of the number of tokens in each query type. The figures contains 14 graphs for the 14 different query types. The x-axis and y-axis presents the number of tokens in the query and their probability density, respectively.