# OpenReview forum: "Complex Logical Reasoning over Knowledge Graphs using Large Language Models"
_ICLR.cc/2024/Conference — ICLR 2024 Conference Withdrawn Submission_

### Official Review · Reviewer_vGc3 · 2023-10-24

**Soundness:** 1 poor
**Presentation:** 3 good
**Contribution:** 1 poor
**Rating:** 3
**Confidence:** 3

**Summary:**

The authors propose a novel neural logic query on Knowledge Graph by using LLMs. A complex logic query is decomposed into simple queries, each will be answered separately by LLM, and then these answers will integrated by following the logical relations among. Experiments were conducted on three benchmark datasets FB15K, FB15K-237, NELL995. Authors claimed the performances have greatly outperformed SOTA level.

**Strengths:**

The paper outlined an exciting application of LLMs to improve neural logic reasoning on knowledge graphs. Figure 1 vividly illustrates the procedure of using query prompts to get single query results from LLM. In most of the experiments, the performances indeed greatly outperform the SOTA level.

**Weaknesses:**

Authors assume that LLMs have the ability to reason and ascribe the improvement of their system's performances to this. However, whether LLMs can reason is still an open question. See below.

C. Biever, The easy intelligence tests that AI chatbots fails, Nature 619 (2023) 686–689
M. Melanie, How do we know how smart AI systems are?, Science 381 (6654) (2023) adj5957

If we look carefully at the experiment results, it is not the case the proposed system consistently greatly outperforms SOTA level. In the FB15K dataset, CQD system outperforms the authors' system in 4 out of 9 tasks. In the NELL995 dataset (in ip task, CQD scores 70.0, while the authors' system scores 29.3), CQD system outperforms the authors' system in 2 out of 9 tasks. In the case of the negation query, BetaE outperforms authors' system in 2 out of 5 tasks in the FB15K datasets.

**Questions:**

Why is the performance of LARK not stable in some datasets but not in others?

Why is the performance of negative queries still much lower than that of positive queries, even supported by LLMs?

In the 3in task, BetaE performs the best in all three datasets, why?

"Our experiments on logical reasoning across standard KG datasets demonstrate that LARK outperforms the previous state-of-the-art approaches by 35% − 84% MRR on 14 FOL". How is 35% − 84% calculated?

---

### Official Review · Reviewer_gvan · 2023-10-30

**Soundness:** 1 poor
**Presentation:** 3 good
**Contribution:** 3 good
**Rating:** 5
**Confidence:** 4

**Summary:**

The paper proposes a novel approach, Language-guided Abstract Reasoning over Knowledge graphs (LARK), for complex reasoning over knowledge graphs (KGs) using large language models (LLMs). The approach decouples KG search and abstract logical query reasoning to leverage the strengths of graph extraction algorithms and LLMs, respectively. The authors demonstrate that their approach outperforms state-of-the-art KG reasoning methods on standard benchmark datasets across several logical query constructs, with significant performance gain for queries of higher complexity.

**Strengths:**

1. Impressive results on NELL dataset. This approach outperforms state-of-the-art KG reasoning methods on standard benchmark datasets across several logical query constructs.
2. Integration of LLMs: The performance of the proposed approach improves proportionally to the increase in size of the underlying LLM, enabling the integration of the latest advancements in LLMs for logical reasoning over KGs.

**Weaknesses:**

1. The example query in Figure 1 equals to 3i. It is not a simplest form. Maybe another query structure is better.
2. RQ3. Only two sizes of LLMs are not sufficient, though we do observe a significant performance improvement on 13B Llama2.
3. The experiment to answer RQ4 is not a good design. GPT3.5 is different from Llama2-7B and Llama2-13B. It’s hard to tell the importance of token limit. Because GPT3.5 outperforms various open-source LLMs (including Llama2-7B and Llama2-13B) on many reasoning benchmarks. A proper way to study how the token limit affects reasoning performance is to vary token limit on one LLM. For example, we could set 100%, 75%, 50%, 25% on max token limit of Llama2.
4. Lack of comparison with other LLM-based approaches: The paper only compares the proposed approach with state-of-the-art KG reasoning methods and does not compare it with other LLM-based approaches for KG reasoning.

**Questions:**

1. The result on NELL dataset is quite impressive. But NELL is constructed from web data, which may have been used for pretraining the LLMs. Could you show that the results from LLM are given by logical reasoning instead of recalling hidden knowledge inside?
2. Integration with other KG reasoning methods: It would be interesting to see how the proposed approach can be integrated with other KG reasoning methods, such as rule-based reasoning or probabilistic reasoning, to further improve the performance.
3. Interpretability: The paper does not discuss the interpretability of the proposed approach, which is an important aspect of KG reasoning. It would be interesting to see how the approach can provide explanations for the reasoning process.

---

### Official Review · Reviewer_5TPZ · 2023-10-31

**Soundness:** 2 fair
**Presentation:** 2 fair
**Contribution:** 2 fair
**Rating:** 5
**Confidence:** 4

**Summary:**

The paper proposes the LARK model that utilizes the reasoning abilities of large language models to efficiently answer FOL queries over knowledge graphs. The LARK model first extracts subgraph contexts and then performs chain reasoning over these contexts. Empirical results on three datasets on FOL query show that the proposed LARK model performs better than other methods that work without large language models.

**Strengths:**

The proposed method is simple yet effective.

The obtained improvement in performance is significant.

The presentations and figures are clear and easy to follow.

The limitations of the proposed method are preliminarily discussed.

**Weaknesses:**

The technical novelty is neutral, as the paper seems to use LLM for FOL queries directly.

The writing of the paper can be largely improved.

The running-time efficiency of LARK is not reported.

The paper is empirically driven and lacks in-depth analysis, whether from methodological or theoretical perspectives.

Besides, the paper does not provide satisfying insights or underlying properties of the LARK model.

As the paper uses entities and relations in queries to find pertinent subgraph contexts, it would be better to discuss some relevant subgraph sampling methods on KG, e.g., AStarNet (Zhaocheng Zhu et al., NeurIPS 2023) and AdaProp (Yongqi Zhang et al., KDD 2023).

**Questions:**

Please refer to the above weakness part.

---

### Official Review · Reviewer_2XTg · 2023-11-09

**Soundness:** 2 fair
**Presentation:** 2 fair
**Contribution:** 2 fair
**Rating:** 3
**Confidence:** 3

**Summary:**

The paper describes a system for executing first-order logic queries on knowledge graphs by using an LLM (Llama-2). The proposed approach:
1. Collects a subgraph around the query entities from the KG
2. Anonymizes the entities and relations in the query and the collected subgraph by converting them into entity and relation ids
2. Prompts the LLM to execute each step of the query graph and passes intermediate answer entities to the next step of execution

- Results on a dataset of complex queries shows improvements over baseline approaches.
- Without any fine-tuning, the LLM is shown to generalize across different underlying knowledge graphs (prior works require at least part of the pipeline to be fine-tuned for each KG)

**Strengths:**

- The proposed approach based on LLMs only observers a subset of the full KG with anonymized entity and relation IDs. This allows the system to:
    - Easily generalize to new KGs
    - Scale to larger KGs (since the LLM only observes a k-hop subgraph)
    - Most existing approaches for KG reasoning need all or part (one-hop link prediction model) to be fine-tuned for each KG
- Authors demonstrate that step-by-step query execution performs significantly better than directly attempting to execute the complex query
- Authors show that other LLMs also seem to handle the task, demonstrating that the performance is not specific to Llama-2

**Weaknesses:**

- The presentation of the paper lacks details of the evaluation protocol
    - The paper evaluates the system on the queries generated by [1]. [1] highlights 2 types of answer entities: trivial answers (query can be directly executed on the available KG to gather answers) and non-trivial answers (queries require reasoning about missing edges).
    - It is unclear if this paper reports results on the trivial, non-trivial, or combined subsets of answers.
- While the paper reports strong results on all benchmarks, there is no indication of the mechanism by which an LLM "could" solve the task.
    - The results are especially surprising given that the LLM only observes anonymized entity and relation names and cannot apply any semantic reasoning
    - The LLM only sees a small subset of the KG, so it does not have sufficient information to learn the semantics of the anonymized relations
- The paper misses some relevant work in query execution on KGs
    - [2] is a strong baseline that uses learned one-hop link prediction models for each KG and uses them to execute complex queries. The method provides guarantees that the trivial answers will be predicted exactly and shows strong performance on the non-trivial answer entities
    - Other competitive baselines include [3] and [4]. [2] provides MRR results for these approaches on the same datasets
    - The MRR for the CQD baseline in this paper do not match the corresponding results in [2]

I have reframed these weaknesses as a series of questions in the next section.

---
[1] Ren, H., Hu, W., and Leskovec, J. Query2box: reasoning over knowledge graphs in vector space using box embeddings. In International Conference on Learning Representations, 2020.

[2] Yushi Bai, Xin Lv, Juanzi Li, and Lei Hou. 2023. Answering complex logical queries on knowledge graphs via query computation tree optimization. In Proceedings of the 40th International Conference on Machine Learning (ICML'23)

[3] Chen, X., Hu, Z., and Sun, Y. Fuzzy logic based logical query answering on knowledge graphs. In Proceedings of the AAAI Conference on Artificial Intelligence, volume 36, pp. 3939–3948, 2022.

[4] Zhu, Z., Galkin, M., Zhang, Z., and Tang, J. Neuralsymbolic models for logical queries on knowledge graphs. In Proceedings of the 39th International Conference on Machine Learning, volume 162, pp. 27454–27478, 2022.

**Questions:**

1. Please clarify the evaluation protocol. Is the reported MRR only considering the non-trivial answer set?
2. What is the baseline MRR of directly executing the queries on the KG? e.g. for the 2p query (e1, r1, ?) -> (?, r2, ANS), explicitly returning entities that are connected to e1 by the 2-hop path (r1, r2)? Let's call this baseline DIRECT
3. What is the MRR of returning the answer set of DIRECT combined with a random order of entities from the collected neighborhood subgraph? Let's call this baseline DIRECT+KNN
4. Given that the LLM only observes anonymized entity and relation IDs and a neighborhood subgraph of the full KG, how does the LLM perform better than other approaches?
    1. The paper is missing a discussion and examples of when and why it performs better than the baselines (including DIRECT and DIRECT+KNN).
    2. In Sec 4.3 the paper claims that the improvement is from "the LLM's ability to capture a broad range of relations...". How is this possible with anonymized relations?
5. Introduction, page 1, para 1: The introduction mentions that the proposed approach handles more complex queries than "constrained" FOL queries. The evaluation datasets in this paper are also FOL queries. How do you support this claim?
    - Same claim made on page 2, para 2
6. Section 3.2, Neighborhood retrieval: There seem to be errors in the equations here. It is unclear how the neighborhood graph actually grows.
    1. Eq (5): By definitions 1-4, $Q_\tau$ only contains the head entity and relation for each edge in the query graph. Then how do you constrain $t \in E^1_r$
     2. Eq (7): Constrains the new graph to contain entities and relations that lie in the previous neighborhood graph. How does the graph ever grow?
     3. When collecting the neighborhood subgraph, do you only include edges that are of the same type as the edges in the query graph?
7. Sec 4.5: The line says "over 4096 and less than 4096". I am assuming this is a typo and you only meant "over 4096"? Please clarify
8. Fig 2: The text of the figure seems to convey that the model receives the logical form of the query as input. Why do you then need a query type identifier according to the description of Fig 2?